# A ST-ConvLSTM Network for 3D Human Keypoint Localization Using MmWave Radar

**DOI:** 10.3390/s25185857

**Published:** 2025-09-19

**Authors:** Siyuan Wei, Huadong Wang, Yi Mo, Dongping Du

**Affiliations:** 1School of Electronic Science and Engineering, Chongqing University of Posts and Telecommunications, Chongqing 400065, China; s240431042@stu.cqupt.edu.cn (S.W.); s220431066@stu.cqupt.edu.cn (Y.M.); 2Chengdu Song Yuan Technology Co., Ltd., Chengdu 618000, China; dudu@micradar.cn

**Keywords:** human keypoint localization, millimeter wave radar, spatiotemporal features fusion, radar point cloud dataset, binocular camera annotation

## Abstract

Accurate human keypoint localization in complex environments demands robust sensing and advanced modeling. In this article, we construct a ST-ConvLSTM network for 3D human keypoint estimation via millimeter-wave radar point clouds. The ST-ConvLSTM network processes multi-channel radar image inputs, generated from multi-frame fused point clouds through parallel pathways. These pathways are engineered to extract rich spatiotemporal features from the sequential radar data. The extracted features are then fused and fed into fully connected layers for direct regression of 3D human keypoint coordinates. In order to achieve better network performance, a mmWave radar 3D human keypoint dataset (MRHKD) is built with a hybrid human motion annotation system (HMAS), in which a binocular camera is used to measure the human keypoint coordinates and a 60 GHz 4T4R radar is used to generate radar point clouds. Experimental results demonstrate that the proposed ST-ConvLSTM, leveraging its unique ability to model temporal dependencies and spatial patterns in radar imagery, achieves MAEs of 0.1075 m, 0.0633 m, and 0.1180 m in the horizontal, vertical, and depth directions. This significant improvement underscores the model’s enhanced posture recognition accuracy and keypoint localization capability in challenging conditions.

## 1. Introduction

The continuous advancement of artificial intelligence, computer vision, and sensor technology has significantly expanded the application landscape for Human Activity Recognition (HAR) systems. Accurately detecting and analyzing human motion poses holds substantial importance across diverse fields, including behavioral monitoring, intelligent security, sports rehabilitation, and human–computer interaction [1,2,3,4]. Although traditionally addressed through computer vision techniques, achieving precise, stable, and real-time extraction of human pose information remains a core challenge. These methods analyze image sequences from monocular or stereo cameras, utilizing RGB, RGB-D, or infrared cameras combined with deep learning to directly classify actions or detect body parts for skeletal inference [5]. Representative approaches include Oxford University’s body-part recognition for posture detection [6], k-poselet agglomerative clustering for multi-person pose estimation [7], and R-CNN-based keypoint masks with ResNet for skeleton reconstruction [8]. The OpenPose framework and datasets from Carnegie Mellon University further established high-precision pose annotation standards [9,10].

However, visible light or infrared camera-based solutions face limitations: they are susceptible to lighting variations, environmental interference, and occlusion, compromising recognition stability and accuracy [11]. Large-scale deployment is also restricted by privacy concerns. These limitations are particularly critical in sensitive application domains such as medical monitoring. For instance, in continuous health assessment tasks like tremor monitoring for Parkinson’s disease patients [12,13], vision-based methods can be hindered by low-light home environments and raise significant privacy issues. In contrast, millimeter-wave (mmWave) radar is recognized as a key sensor for next-generation intelligent perception because it is insensitive to ambient light, with strong penetration capability and high measurement precision [14,15]. Crucially, it provides inherent privacy preservation by capturing motion data without identifying facial or body features, thus aligning with ethical guidelines for long-term patient monitoring, and it can determine whether the patient’s condition is abnormal through posture detection. Its robustness to occlusion further enhances its suitability for such use cases. When applied to HAR, mmWave radar emits Frequency-Modulated Continuous Wave (FMCW) signals and analyzes echoes to generate 3D spatial point clouds of human body scattering points, enabling motion pose detection [16,17,18,19,20].

Recent research has demonstrated mmWave radar’s efficacy in skeletal tracking. MIT researchers pioneered a method using FMCW radar to capture human point clouds, applying deep learning models to detect and track skeletal keypoints. While effective in multi-person scenarios, their approach involves relatively simplistic point cloud processing and requires robustness improvements in complex environments [21]. Similarly, a UCLA team proposed a Convolutional Neural Network (CNN) to extract features from point cloud data, integrating spatial information for keypoint detection and tracking, which also achieved promising multi-person results [22]. These advancements highlight mmWave radar’s potential to overcome traditional vision limitations, though challenges in point cloud processing precision and environmental adaptability remain active research areas.

To overcome the limitations in current estimation of 3D skeletal keypoint coordinates from mmWave radar point clouds, we propose a comprehensive solution for human pose recognition using radar point clouds. Firstly, we design a dedicated stereo-camera-based human motion pose data testing system, by which a new mmWave radar 3D human keypoint dataset (MRHKD) is built for deep network training. Secondly, a novel ST-ConvLSTM network model is specifically designed for regressing skeletal keypoints from sparse point clouds. This integrated approach aims to achieve more stable motion pose recognition performance in challenging scenarios characterized by high noise levels and diverse postures. Before the model training process, point cloud data have been clustered and fused to improve the training performance of the ST-ConvLSTM. Finally, the performance of the ST-ConvLSTM is evaluated.

The paper is organized as follows. Section 2 provides the related work in the field. Section 3 details the generation of the dataset MRHKD. Section 4 introduces radar signal processing. The structure of the ST-ConvLSTM model is elaborated upon in Section 5. Section 6 presents the experimental results. Finally, the study is concluded in Section 7.

## 2. Related Works

In the field of Human Activity Recognition, pose estimation serves as a core technology for inferring behavioral intent, with implementation approaches primarily divided into vision-based sensors and radio frequency (RF)-based sensors. Vision-based methods employ monocular cameras, RGB-D cameras, or infrared cameras combined with deep learning algorithms to directly classify human actions or infer skeletal poses by detecting body parts. However, monocular systems struggle to obtain reliable depth information. To address this limitation, the University of Toronto team developed the HumanEva dataset, which utilizes seven synchronized cameras (three RGB + four grayscale) in a circular array with reflective markers placed on body joints, leveraging the commercial ViconPeak system to capture ground-truth 3D poses [23]. Another representative solution, Microsoft Kinect, integrates RGB and infrared cameras for 3D scene capture [24]. Nevertheless, vision-based methods remain inherently susceptible to lighting variations, occlusions, and privacy concerns in 3D estimation.

In contrast, RF sensors such as mmWave radar detect targets using self-emitted signals, offering strong resistance to ambient light interference and inherent privacy advantages. The technological evolution traces back to 2003 when an MIT team achieved human gait monitoring using Ultra-Wideband (UWB) sensors, marking RF technology’s initial application in motion analysis [25]. Early research focused on action classification: Young et al. collected data from 12 subjects performing seven activities using Doppler radar, extracting six features from time-varying Doppler images, achieving nearly 90% detection accuracy with Artificial Neural Networks (ANNs) and Support Vector Machines (SVMs) [26]; Cao et al. employed Deep Convolutional Neural Networks (DCNNs) for individual and group walking gait classification, demonstrating significantly superior performance over traditional supervised classifiers like Bayesian methods [27]; and, to reduce annotation costs, Li proposed the semi-supervised transfer learning algorithm “Joint Domain and Semantic Transfer Learning (JDS-TL)”, utilizing sparsely labeled datasets to alleviate the burden of large-scale radar signal annotation [28].

Recently, RF-based skeletal tracking has emerged as a new research direction, primarily involving two data processing paradigms, RF heatmaps and RF point clouds. For heatmap processing, MIT’s 2015 RF-Capture used FMCW signals and antenna arrays to reconstruct poses by identifying and stitching body part contours, though with limited temporal tracking capability [29]. The 2018 RF-Pose innovatively employed dual horizontal and vertical antenna arrays to capture heatmap data, combining a “teacher–student” learning framework with an encoder–decoder network for keypoint prediction [30]. Further advancement appeared in RF-based 3D Skeletons, which utilized 1.8 GHz bandwidth FMCW signals and ResNet architecture for 3D keypoint estimation, reconstructing skeletal models via triangulation while using OpenPose-provided visual skeletal data for supervised training [31]. For point cloud processing, the high-dimensional, sparse nature of point cloud data poses dual challenges for deep learning models—requiring substantial computational power for processing and strong generalization capabilities due to annotation difficulties. Addressing these, Yu et al. constructed a benchmark radar point cloud dataset for human activities, employing DBSCAN clustering for point cloud segmentation and Long Short-Term Memory (LSTM) networks for classification, achieving more than 95% accuracy across four action classes [32]. Li Zhe-yuan proposed an improved DBSCAN algorithm that integrates density-based and partition-based clustering advantages to reduce computational complexity, combined with Extended Kalman Filters and Joint Probabilistic Data Association for high-precision multi-target tracking.

Despite significant progress in millimeter-wave radar point cloud technology for pose estimation, three core challenges persist: sparse point clouds hindering per-frame feature extraction, high costs of 3D keypoint annotation, and existing models’ difficulties in balancing computational efficiency with spatiotemporal modeling capabilities. Future research necessitates deeper integration of advanced deep learning techniques to develop efficient point cloud processing methods and robust keypoint regression architectures, thereby advancing real-time pose tracking in complex scenarios.

## 3. Building Dataset

### 3.1. Dataset Definition and Design

Current datasets such as HumanEva and PNHM mainly discuss millimeter-wave radar data at the behavioral or action classification level, with limited exploration of 3D positioning or pose recognition at the joint level [23,33]. This gap necessitates a dedicated radar point cloud dataset annotated with human keypoint coordinates. By providing high-precision 3D annotations of human keypoints combined with radar point clouds and motion information, we constructed the mmWave radar human keypoint dataset (MRHKD), which comprises approximately 73,794 frames of data captured from four subjects. The data collection was conducted in diverse environments, including both indoor and outdoor settings, to incorporate variability in background clutter, lighting conditions, and multipath interference. The recorded poses encompass a wide range of human motions such as standing, walking forward and backward, turning left and right, raising arms, and leaning left and right.

In the deep learning model adopted for this study, accurate regression of 3D coordinates or motion trajectories for each joint requires supervision signals from annotations that are both precise and consistent with radar scattering characteristics. Consequently, the human motion pose dataset designed herein incorporates the following elements: (1) 3D coordinates of 12 keypoints (head center, left ear, right ear, torso center, left shoulder, right shoulder, left hip, right hip, left elbow, right elbow, left knee, and right knee), with all (*x*, *y*, *z*) coordinates annotated in a unified world coordinate system where *x* represents the horizontal direction, *y* represents the vertical direction, and *z* represents the depth direction; and (2) radar point cloud data matrix P with additional attributes, generated through radar echo processing. Each scatter point contains (*x*, *y*, *z*) coordinates, along with velocity, signal-to-noise ratio (SNR), and a motion category, including dynamic, sustained micro-motion, and brief micro-motion.

To enable efficient loading and matching of keypoints with scatter points during training and inference, we implement a unified recording format. For each timestamp t, point cloud data and keypoint coordinate matrix K are aligned with saved files and saved in the same records, as exemplified in Table 1. Each frame contains 12 keypoint coordinates, N point cloud positions, per-point velocity or SNR, a motion category or confidence values, and a timestamp. Here, N varies per frame based on point density.

Regarding dataset partitioning for model training, this study divides the available data into three subsets. Specifically, 70% of the training set is used for model learning, 20% of the test set is used for periodic model evaluation during training, while the remaining 10% of the validation set is specifically used for final performance evaluation and parameter selection verification after training. Crucially, this split is performed at the subject level, meaning that all data from any single subject is exclusively allocated to only one of the three subsets. This subject-exclusive partitioning strategy helps prevent overfitting to subject-specific characteristics and provides a more rigorous and generalizable evaluation of the model’s performance.

### 3.2. Hardware Design of Dataset Testing System

Accurately capturing 3D coordinates is essential for building human pose datasets. To enhance the performance of subsequent neural network models, we utilized the Hybrid Human Motion Annotation System (HMAS) to construct a millimeter-wave radar three-dimensional human keypoint dataset for network training. The hardware configuration of the radar point cloud HMAS includes a FPGA board, paired with two cameras of model OV5640 to form a dual-camera module. This system uses binocular cameras, which provide sufficient skeletal keypoint accuracy within 2.5–5 m through stereo vision. The hardware connection and data flow path of the experimental platform are shown in Figure 1.

As depicted in Figure 2, the HMAS generates synchronized datasets through parallel radar and camera processing. Radar-side processing is accomplished through the radar processing chip module by completing FFT range–Doppler analysis and Constant False Alarm Rate (CFAR) algorithms to generate the initial point cloud with adaptive thresholds. These initial point clouds are refined via an enhanced clustering algorithm to remove noise and segment targets, yielding clean point clouds with velocity or confidence data. Then, the processed point cloud data is fused into multiple frames through the Iterative Closest Point algorithm based on the latest iteration, which is shown in Section 4.

Camera-side processing uses the FPGA for image preprocessing before camera calibration in MATLAB R2020a. Through nonlinear optimization such as Zhang Zhengyou’s calibration method and the least square method, the internal parameters, external parameters, and distortion parameters of the camera can be obtained [34]. Based on the internal and external parameter information, OpenCV can be used to perform distortion correction and polar line alignment on the image. Then, the MoveNet model is used to identify the keypoints of the human body [35]. Finally, the three-dimensional coordinates are reconstructed by combining the calibrated internal and external parameters of the binocular system, as introduced in Section 3.3.

Temporal synchronization is achieved by triggering both sensors simultaneously from the host and compensating for processing delays via FPGA timestamps, with details provided in Section 3.4. It produces time-aligned radar point clouds with associated velocity and confidence metrics and corresponding 3D skeletal keypoints, forming a unified dataset for human motion analysis and model training or evaluation.

### 3.3. Keypoint 3D Coordinate Computation

Binocular vision systems reconstruct 3D scenes by simulating human stereoscopic disparity, with core processes encompassing coordinate system definition, projection modeling, depth calculation, and 3D coordinate resolution. Firstly, the system involves three types of coordinate systems, the image coordinate system $\left(u,v\right)$, a 2D pixel-based plane; the camera coordinate system ($x_{c},y_{c},z_{c}$), with its origin at the camera’s optical center and the optical axis as $z_{c}$; and the world coordinate system ($x_{w},y_{w},z_{w}$), which acts as the global reference frame.

In binocular stereo vision, two cameras with a fixed-baseline distance b simultaneously capture images of the same scene. By comparing the two images, the depth information of objects in the scene can be extracted. The binocular imaging model is constructed as shown in Figure 3. In the world coordinate system, the optical axis direction of the left camera can be defined as the z-axis, the horizontal direction (baseline direction) as the x-axis, and the vertical direction as the y-axis. The right camera is translated by b along the x-axis. Let there be a point $P_{w}(x_{w},y_{w},z_{w})$ in space, whose projected pixel coordinates in camera 1 and camera 2 are ${(u}_{L},v_{L})\;$ and ${(u}_{R},v_{R})$, respectively. Let $x_{left}$ represent the horizontal coordinate of the 3D point on the left image, and $x_{right\;}$ represent the horizontal coordinate on the right image.

According to the principle of similar triangles,(1)$$\frac{z}{f_{piexl}}=\frac{x}{x_{left}}=\frac{x-b}{x_{right}}$$
where $f_{pixel}\;$ represents the camera’s pixel focal length, obtained by dividing the camera’s optical focal length by the size of a single pixel. Rearranging the above equation yields the depth z as(2)$$z=\frac{f_{pixel}\cdot b}{x_{left}-x_{right}}=f\cdot \frac{b}{d}$$
where d represents disparity value between camera 1 and camera 2. Here, $\;z_{c}=z\;$.

According to Zhang Zhengyou’s calibration method [22], the mathematical relationships between the coordinates in the 2D image coordinate system, the 3D camera coordinate system, and the 3D world coordinate system are given by(3)$$z_{c}\left[\begin{array}{c} u\\ v\\ 1 \end{array}\right]=K\cdot \left[R\left|T\right.\right]\cdot \left[\begin{array}{c} \begin{array}{c} x_{w}\\ y_{w}\\ z_{w} \end{array}\\ 1 \end{array}\right]$$
where K  is the intrinsic matrix, which characterizes the camera’s projection model, and it can be expressed as Equation (4), and $\left[R\left|T\right.\right]$ represents the extrinsic parameters, which describe the position and orientation of the camera relative to the world coordinate system.(4)$$K=\left[\begin{array}{ccc} f_{x} & s & u_{0}\\ 0 & f_{y} & v_{0}\\ 0 & 0 & 1 \end{array}\right]$$
where $f_{x},f_{y\;}$ represent the focal lengths in the horizontal and vertical directions, respectively, and $u_{0},v_{0}\;$ represent the intersection point of the optical axis with the image plane.

The extrinsic matrix is composed of a rotation matrix R and a translation vector T, which is commonly denoted as Equation (5), where $r_{ij}\;$ is the rotation coefficient of the rotation matrix, and $t_{k}\;$ is the translation coefficient of the translation vector.(5)$$\left[R\left|T\right.\right]=\left[\begin{array}{cccc} r_{11} & r_{12} & r_{13} & t_{1}\\ r_{21} & r_{22} & r_{23} & t_{2}\\ r_{31} & r_{32} & r_{33} & t_{3} \end{array}\right]$$

Since the origin of the world coordinate system is set at the center of the calibration board’s plane, for the extrinsic matrix at a specific depth position $z_{c}$, we can set $z_{w}=$ 0. Thus, Equation (3) simplifies to(6)$$z_{c}=K\cdot \left[\begin{array}{ccc} r_{11} & r_{12} & t_{1}\\ r_{21} & r_{22} & t_{2}\\ r_{31} & r_{32} & t_{3} \end{array}\right]\cdot \left[\begin{array}{c} x_{w}\\ y_{w}\\ 1 \end{array}\right]$$

Let(7)$$A=K\cdot \left[\begin{array}{ccc} r_{11} & r_{12} & t_{1}\\ r_{21} & r_{22} & t_{2}\\ r_{31} & r_{32} & t_{3} \end{array}\right]$$

Then, when the intrinsic and extrinsic matrices, depth coordinate, and image pixel coordinates are known, we can obtain the following coordinates:(8)$$\left[\begin{array}{c} x_{w}\\ y_{w}\\ 1 \end{array}\right]=A^{-1}\cdot z_{c}\cdot \left[\begin{array}{c} u\\ v\\ 1 \end{array}\right]$$

### 3.4. Kalman Filter Algorithm

To achieve stable and smooth spatial coordinate estimation, a Kalman Filter can be applied temporally [36]. The Kalman Filter combines measurement sequences with a system motion model to suppress transient noise and automatically interpolate missing data, thereby enhancing stability in coordinate and velocity estimation. The algorithm defines the state vector $x_{t}={\left[p_{t},v_{t}\right]}^{T}$, where $\;p_{t}$ represents the keypoint coordinate, and $v_{t}$ is the corresponding velocity component. The system assumes that short-term motion follows a constant-velocity model, with the state transition equation:(9)$$x_{t+1}=\left[\begin{array}{c} p_{t+1}\\ v_{t+1} \end{array}\right]=F\cdot x_{t}+w_{t}=\left[\begin{array}{cc} 1 & ∆t\\ 0 & 1 \end{array}\right]x_{t}+w_{t}$$

Here, F is the state transition matrix, ∆t is the inter-frame time interval, and $w_{t}$ is process noise. The observation model directly links to sensor measurements:(10)$$z_{t}=H\cdot x_{t}+r_{t}=\left[\begin{array}{cc} 1 & 0\\ 0 & 1 \end{array}\right]x_{t}+r_{t}$$
where H is the observation matrix and $r_{t}$ is measurement noise. The Kalman filter iterates through prediction and update stages. In the prediction stage, it computes the prior state and covariance using the previous posterior estimate:(11)$$x_{t\left|t-1\right.}=F\cdot x_{t-1\left|t-1\right.}\;,\;P_{t\left|t-1\right.}=FP_{t-1\left|t-1\right.}F^{T}+Q$$
where Q is the process noise covariance, $x_{t\left|t-1\right.}$ represents the predicted state mean, and $P_{t\left|t-1\right.}$ represents the predicted covariance. In the next stage, it incorporates new observations $z_{t}$, calculates the Kalman gain $K_{t}$, and updates the stage:(12)$$K_{t}=P_{t\left|t-1\right.}H^{T}{\left(HP_{t\left|t-1\right.}H^{T}+R\right)}^{-1}$$(13)$$x_{t\left|t\right.}=x_{t\left|t-1\right.}+K_{t}\left(z_{t}-Hx_{t\left|t-1\right.}\right),\;P_{t\left|t\right.}=\left(I-K_{t}H\right)P_{t\left|t-1\right.}$$

The gain $K_{t}$ dynamically balances the reliability of predictions versus observations: increasing measurement noise R shifts reliance toward the motion model, while increasing process noise Q favors sensor data. Figure 4 displays the data curves of a human subject moving back and forth along the depth direction before and after applying the Kalman filtering algorithm. As observed in the figure, the curve labeled Raw is before applying the Kalman filtering algorithm, the curve labeled Optimize is after applying the Kalman filtering algorithm. Whether it is the nose, the left eye, or the right eye, the Kalman filter demonstrates significant smoothing effects on the data, effectively suppressing fluctuations. Furthermore, it successfully predicts outcomes for frames with missing values.

### 3.5. Synchronization of Radar and Vision Systems

To unify the radar and camera coordinate systems, a reflective reference object is captured by both sensors. The radar detects its scattering center $p_{r}$, while stereo cameras compute its 3D center $p_{c}$. Both satisfy the rigid transformation model,(14)$$p_{r}=R\cdot p_{c}+t$$
where R is a rotation matrix of 3 × 3, and t is a translation vector of 3 × 1. R and t can be solved by least squares or direct alignment formulas.

Due to its long processing chain, radar data inherently lags behind camera data when the host sends UART commands for radar acquisition and FPGA processing. To synchronize, the host sends simultaneous start signals to both sensors. An FPGA timer measures the radar’s processing delay ∆t, primarily from signal transmission. This delay corrects radar timestamps during fusion. Camera frames via UDP and adjusted radar frames thus correspond to identical motion instants, enabling precise multimodal analysis.

## 4. Radar Signal Processing Flow

### 4.1. Point Cloud Clustering Algorithm

Millimeter-wave radar-generated point cloud data often exhibits significant sparsity and noise. Therefore, clustering algorithms are typically employed for point cloud segmentation and feature extraction. To balance flexible updates and density adaptivity, this study proposes an improved DBSCAN clustering algorithm, Adaptive DBSCAN (A-DBSCAN), that integrates partitioning and density concepts. Specifically, the algorithm begins by selecting an unlabeled point as the initial cluster center, dynamically assigns neighboring points, and updates the centroid position in real time. During the assignment process, cluster stability is evaluated against a minimum point threshold (MinPts): if the current cluster’s point count falls below this threshold, it is marked as a transitional cluster; otherwise, it is designated as a stable cluster. When a point’s distance to all existing cluster centers exceeds a dynamic threshold, a new cluster creation mechanism is triggered. The iterative process continues until cluster center movement drops below a convergence threshold or the maximum iteration count is reached, ultimately outputting valid clusters satisfying MinPts and noise points, as shown in Figure 5.

The algorithm’s innovation lies in its dynamic parameter design and performance advantages. For parameter adaptation, (1) the neighborhood radius ε is dynamically determined by identifying the maximum curvature point in the sorted nearest-neighbor distance distribution, eliminating manual tuning. (2) The MinPts threshold is dynamically adjusted based on local density: it is increased in high-density regions to suppress over-segmentation and decreased in low-density areas to enhance sensitivity. The proposed fusion approach retains K-means’ efficient centroid-updating characteristics while inheriting DBSCAN’s adaptability to noise and complex-shaped clusters, significantly overcoming limitations of traditional clustering algorithms in sparse point cloud scenarios.

### 4.2. Point Cloud Fusion Algorithm

In traditional mmWave radar signal processing, CFAR algorithms employ high thresholds to reduce false alarms. This approach filters out weaker targets, resulting in sparse point cloud images where a single frame typically contains only tens to hundreds of points. To address the issues of sparsity in single-frame point clouds and dynamic interference, multi-frame point cloud fusion is typically employed. This approach significantly enhances perception quality by integrating spatiotemporal information. Current research on multi-frame point cloud fusion focuses on algorithmic innovation, efficiency optimization, and scene adaptability [37,38,39]. This study proposes a multi-frame point cloud fusion algorithm based on the Iterative Closest Point (ICP) method. The ICP algorithm achieves fusion by matching corresponding point pairs between a source point cloud $p_{s}^{i}$ and target point cloud $p_{t}^{i}$. It computes an optimal 2D rigid transformation (rotation matrix R* and translation vector t*) using point-to-point constraints. This transformation aligns $p_{s}^{i}$ with the $p_{t}^{i}$ coordinate system, effectively merging the clouds and enhancing point density. The most recent iteration point algorithm can be described by the following formula:(15)$$R*,t*=argmin\frac{1}{N}\sum_{i=1}^{N}{\left\|p_{t}^{i}-\left(R\cdot p_{s}^{i}+t\right)\right\|}^{2}$$

Here, N is the number of matched points, $p_{s}^{i}$ and $p_{t}^{i}$ denote corresponding points in the source point cloud and target point cloud, and R* and t* represent the optimal transformation matrices. The solution of Equation (15) is achieved through three iterative steps: nearest-neighbor matching, optimal rotation matrix estimation, and optimal translation matrix estimation, with the aim of multi-frame fusion. For nearest-neighbor matching, the commonly used method is to calculate the nearest point based on the Euclidean distance. This step usually requires an efficient nearest neighbor search algorithm. By sequentially applying the ICP to consecutive radar frames and integrating results through a sliding window approach, the fused multi-frame point cloud replaces sparse single-frame data, significantly increasing point density per output frame.

## 5. ST-ConvLSTM

### 5.1. Overall Architecture of ST-ConvLSTM

Directly inputting high-dimensional, sparse 3D millimeter-wave radar point clouds into traditional CNNs is complex and fails to leverage the strengths of 2D convolution kernels. To address this, this study transforms the raw point cloud into 2D image-like data through projection and channel conversion. This processed data is then fed into a CNN-based neural network model, the ST-ConvLSTM, designed for human motion pose recognition. The ST-ConvLSTM features two separate inputs, each representing a distinct coordinate projection. Simply merging these projections into a single input risks channel confusion and prevents specialized feature extraction. Therefore, the ST-ConvLSTM uses two parallel CNN branches, one for each projection, which is similar to the mmPose model [21]. The feature maps from both branches are then concatenated along the channel dimension. This combined representation feeds into subsequent fully connected layers, enabling the joint learning of integrated features from both projections. The basic structure of the ST-ConvLSTM is illustrated in Figure 6.

To reduce model parameters and computational load, we employ inverted residual (IR) bottleneck blocks from MobileNetV2 [40]. These blocks utilize depthwise separable convolutions in an inverted structure for efficiency. Between convolutional layers, we insert ConvLSTM cells to model temporal relationships across frames [41]. Given the computational cost of bottleneck LSTM scales with input size, we apply a stride of two in the first IR block to reduce dimensionality. Finally, we refine spatiotemporal features from the bottleneck LSTMs using three additional IR blocks. For temporal feature extraction, we propose a time-distributed CNN and bidirectional LSTM architecture. It consists of four time-distributed convolutional modules, each with batch normalization, one global average pooling layer, and then one concatenate layer to concatenate two branches, two dense layers, a bidirectional LSTM layer, and an output layer. The bidirectional LSTM processes sequences in both directions: one layer operates on the original input while the other uses a reversed copy preserving contextual information from past and future states. Finally, a reverse normalization of coordinates is performed. Since the labels were normalized to [0, 1] or [0, 255] during the training phase, the predicted outputs need to be inversely mapped back to the real-world coordinate range after inference to obtain actual physical distances.

### 5.2. Data Preprocessing

To leverage mature 2D convolutional neural networks, CNNs, for feature extraction, this study converts 3D radar point cloud data (an N × 5 matrix containing *x*, *y*, *z* coordinates, velocity, and confidence per point) into 2D image-like structures. The point cloud is projected onto two orthogonal planes: the xoy plane and the xoz plane. For each projection, two spatial coordinates are mapped to the red and green channels of an image. The third channel, Blue, encodes either velocity or confidence values. This creates two separate H × W × 3 images, where H and W define the image dimensions, and each pixel represents a projected point, as shown in Figure 7.

To fit the 8-bit RGB range (0–255), the point cloud data undergoes normalization. In this study, the 3D coordinates (*x*, *y*, *z*) of the point cloud fall within [0, 5] m, velocity ranges from [−5, 5] m/s, and confidence scores are within [0, 1]. We therefore project these point clouds attribute into the [0, 255] range. The coordinate normalization is implemented using Equation (16):(16)$$\begin{array}{c} x^{\prime }=round\left(\frac{x}{5}\times 255\right)\\ y^{\prime }=round\left(\frac{y}{5}\times 255\right)\\ z^{\prime }=round\left(\frac{z}{5}\times 255\right) \end{array}$$

The Blue channel confidence level or speed can be determined whether to be normalized based on actual needs. Confidence values within [0, 1] can be directly multiplied by 255 for 8-bit precision. Velocity normalization, if required, is achieved using Equation (17):(17)$$v^{\prime }=round\left(\frac{v+5}{10}\times 255\right)$$

Since the number of points per frame varies, unused pixels in the fixed-size H × W image are filled with zeros to ensure consistent input dimensions for the CNN. These processed images are then fed into parallel CNN branches for feature extraction.

## 6. Experiments and Results

### 6.1. Experiment Platform

In the experimental scenario, this study captured binocular image sequences containing human subjects and processed the image sequences using the aforementioned HMAS and algorithm to obtain the 3D coordinates of the human body. Figure 8a displays the physical setup of the HMAS for data acquisition in the scene.

The experimental platform adopts the 60 GHz millimeter-wave radar module provided by Chengdu Song Yuan Technology Co., Ltd. (Chengdu, China). as the core hardware. The millimeter-wave radar module is shown in Figure 8b, configured with 4 transmitting antennas and 4 receiving antennas to enable multi-angle detection capabilities. The radar signal process is completed with the processor in the module and the point cloud data is sent to host through UART port at rate of 5 frames per second. During the development and experimental phases of ST-ConvLSTM, the computing hardware configuration employed for conducting training is detailed in Table 2. The software environment employed the deep learning frameworks and numerical libraries listed in Table 3.

The proposed system offers significant potential for practical deployment and adaptation to new environments. The HMAS, including the computing unit, radar module, and binocular, is compact and can be easily reassembled into a new environment. Although a 60 GHz radar is used in this study, the architecture is compatible with various radar specifications, allowing the use of lower-cost hardware without major losses in accuracy, as long as basic point cloud output and sufficient resolution are maintained. Scaling to multiple sites may face challenges such as initial stereo camera calibration and maintaining consistent lighting conditions. However, the radar’s robustness to ambient light reduces environmental dependencies. Computationally, while model training requires GPU resources, real-time inference can run efficiently on moderate hardware, supporting broader application.

### 6.2. Model Training

The ST-ConvLSTM outputs the 3D coordinates $\left({\hat{x}}_{i},{\hat{y}}_{i},{\hat{z}}_{i}\right)$ of 12 keypoints, where i  denotes the keypoint index. The output matrix is one-dimensional, resulting in an output shape of (1, 12 × 3). An inverse transformation remaps the network’s regression results from the [0, 255] range back to real-world coordinates.

For model training, we employed the following configurations: batch size was set to 24, which enables more sample averaging during each forward–backward propagation, resulting in smoother gradient updates while balancing training speed and computational resource consumption; the Adam optimizer with an initial learning rate of 0.001 ensures rapid loss reduction during early training stages without causing excessively volatile updates; and training progress is monitored through validation or test set loss curves to determine learning rate decay. A learning rate decay factor of 0.8 was triggered if the validation loss plateaued for three consecutive epochs, facilitating model fine-tuning and preventing excessive oscillation. During training, each epoch consists of one full iteration over the entire training dataset. After every epoch, the model is evaluated on the test set. The test error or metric from the current epoch is compared with that of the previous epoch. If a significant improvement is observed, the current learning rate and hyperparameters are maintained. Otherwise, if no improvement is seen for an extended period, strategies like adjusting the learning rate schedule or triggering early stopping are implemented.

Throughout all training iterations, the loss function (loss) and Mean Absolute Error (MAE) were employed as evaluation metrics. MAE is a common evaluation metric for regression tasks, providing an intuitive reflection of the model’s average error magnitude when predicting target values. A smaller MAE indicates more accurate model predictions. The calculation formula for MAE is shown in Equation (18):(18)$$MAE=\frac{1}{N}\sum_{i=1}^{N}\left|p_{i}-{\hat{p}}_{i}\right|$$

Here, N is the total number of samples, $p_{i}$ denotes the true target value of the *i*-th sample, and ${\hat{p}}_{i}$ represents the predicted target value of the *i*-th sample by the model.

By observing the loss and MAE curves, one can intuitively understand the model’s convergence speed, stability, and generalization capability during training. This analysis provides valuable reference for subsequent model selection, tuning, and deployment. The loss value curves and MAE curves for all four networks throughout the training period are plotted in Figure 9. The curve labeled Train Loss represents the loss value curve on the training set, while Validation Loss denotes the loss value curve on the validation set. The curve labeled Train MAE represents the MAE curve on the training set, while Validation MAE represents the MAE curve on the validation set.

Typically, training spans several dozen epochs, depending on the dataset size and model complexity. For the ST-ConvLSTM, a relatively significant decrease in loss is usually observable within the first 10–20 epochs. After a further 20–30 epochs of refinement, the model often reaches a stable convergence region. Additionally, as shown in Figure 9, both the loss and MAE of this network are small, and they remain at a low level and tend to stabilize with the training epochs.

### 6.3. Model Performance Evaluation

To evaluate and visually demonstrate the ST-ConvLSTM’s prediction effectiveness for human keypoints across different time frames, the experiment selected four frames at varying depths *z*, as shown in Figure 10. Each selected frame is split into two subfigures: the radar point cloud overlaid with model-predicted keypoints, and the corresponding camera image.

The figure illustrates the ST-ConvLSTM performing 3D inference on human keypoints at different distances approximately of 1.5 m, 2.5 m, 3 m, and 4 m, superimposing them onto the radar point cloud for comparison with the visible light image. The model-predicted joints connected by skeletal lines exhibit a distribution conforming to the human structure on the point clouds across all frames and generally align with the poses observed in the camera images. In the frames at closer distances of 1.5 m and 2.5 m, the point clouds display more distinct shoulder and torso shapes within the 3D coordinates. The joint positions predicted by the model align closely with the actual limb positions observed in the visible light images. In contrast, at distances of 3 m and 4 m, the human body experiences greater impact from radar scattering intensity and viewing angle. Overall, this model can accurately reflect the keypoints of the human body at different distances.

We conducted a comprehensive ablation study to evaluate the contribution of ICP multi-frame fusion. As Table 4 shows, using only single-frame point clouds yields an MAE of 0.0274 m and MAD of 0.0258 m. With ICP multi-frame fusion, the MAE is reduced to 0.0115 m and the MAD to 0.0102 m—corresponding to a reduction in MAE of approximately 58.0% and a decrease in MAD of about 60.5%. This underscores the importance of temporal integration in mitigating sparsity and instability in single-frame radar point clouds. By merging consecutive frames, the ICP enhances spatial consistency and point density, leading to more stable feature representations and significantly improved robustness.

To quantitatively evaluate model performance, the localization differences between the model-predicted 3D keypoint coordinates and the ground-truth 3D keypoint coordinates obtained via the annotation system were calculated on the test set. MAE and Median Absolute Deviation (MAD) were used as evaluation metrics, and frame-error curves for the three axes (*x*, *y*, *z*) were plotted. And we compared the proposed network model, the ST-ConvLSTM, with the baseline models mmPose [21], MnPoTr [42], and M^4^esh [43]. The network architectures of mmPose, MnPoTr, and M^4^esh were reproduced and validated on the test set. Figure 11 plots the frame-by-frame errors along the *x* (horizontal), *y* (vertical), and *z* (depth) axes for all evaluated models.

As seen in Figure 11, there are fluctuations in the error between the keypoints predicted by the ST-ConvLSTM model and the ground-truth keypoints in the dataset along the *x*, *y*, and *z* axes. However, the error range remains at a relatively low level for most of the time, indicating that the model outputs exhibit no significant extreme mismatches or large outliers in the vast majority of frames. The occurrence of distinct transient spikes in individual frames suggests possible misjudgments of certain joint positions within those specific frames. The stability of error distribution, as illustrated by the MAD and MAE curves in Figure 11, further validates the superiority of the ST-ConvLSTM. Across the horizontal, vertical, and depth axes, the ST-ConvLSTM consistently achieves the lowest error curves. In contrast, the curves of mmPose, MnPoTr, and M^4^esh not only exhibit higher overall error levels but also show significant variability. Particularly in the depth direction, the error curves of MnPoTr and M^4^esh deviate substantially from that of the ST-ConvLSTM, highlighting their severe deficiencies in depth estimation. Although mmPose outperforms MnPoTr and M^4^esh, its errors in the vertical and depth directions remain notably higher than those of the ST-ConvLSTM. This stability stems from the ST-ConvLSTM’s effective capture of spatio-temporal features in radar point clouds, enabling more reliable keypoint localization in complex scenarios.

To further analyze the precision performance and error distribution of keypoint localization across different axes, the mean, range, and median of the model’s data in Figure 11 were calculated based on the MAE curve, with results presented in Table 5. In the horizontal direction, the ST-ConvLSTM model exhibited an overall mean error of approximately 10 cm, with a brief deviation reaching about 19 cm under the most extreme conditions. The close numerical values of the mean and median indicate that most frames did not contain extreme outliers, suggesting a relatively concentrated error distribution.

The average errors of the ST-ConvLSTM in the horizontal, vertical, and depth directions are 0.1075 m, 0.0633 m, and 0.1180 m, respectively. The performance in all three directions is lower than that of mmPose, MnPoTr, and M^4^esh, and the depth error is significantly lower than that of MnPoTr and M^4^esh. Crucially, the depth error range of the ST-ConvLSTM is 0.2809 m, significantly smaller than mmPose’s 0.9646 m. Moreover, the difference between its mean and median is minimal, further demonstrating the predictive robustness against sparse point clouds and environmental disturbances. In contrast, mmPose suffers a vertical drift of about 0.4 m, while MnPoTr and M^4^esh exhibit depth estimation failures, exposing fundamental deficiencies in spatiotemporal feature modeling. In stark contrast, the ST-ConvLSTM maintains the narrowest error bounds (0.1995 m in the horizontal direction; 0.1945 m in the vertical direction) through integrated spatiotemporal convolutions and memory mechanisms. Its significant reduction in maximum depth deviation versus mmPose demonstrates resistance to occlusion, noise, and point cloud sparsity, solidifying its superior performance in millimeter-wave pose estimation.

MAE was computed on the test set after each epoch and used as an evaluation metric. And the additional evaluation of loss and MAE was performed on an independent validation set to ensure the model’s generalization performance on unseen data, as summarized in Table 6. According to the test set results in Table 5, the ST-ConvLSTM achieves a loss value of 2.6443 × 10^−4^ and an MAE of 0.0115, both of which are substantially lower than those of mmPose, MnPoTr, and M^4^esh networks. Specifically, mmPose records a loss of 8.3952 × 10^−4^ and an MAE of 0.0191, while MnPoTr and M^4^esh exhibit even higher metrics. These results indicate that the ST-ConvLSTM yields smaller overall prediction errors and better model fitting in the regression task of 3D human keypoint coordinates.

In summary, ST-ConvLSTM achieves remarkable accuracy in 3D human keypoint localization. MAE results on the test dataset indicate that the ST-ConvLSTM provides keypoint coordinate predictions closely aligned with ground truth in the majority of frames, significantly outperforming baseline models.

## 7. Conclusions

This paper presents a ST-ConvLSTM network for 3D human keypoint localization, which can extract spatiotemporal features crucial for skeletal keypoint identification directly from mmWave radar point clouds. To achieve better network performance, we built a test system specifically designed for 3D human keypoint positioning, the HMAS. Moreover, through this test system, a dedicated dataset, the MRHKD generation framework, has also been implemented to provide training data. Experimental results demonstrate that the proposed ST-ConvLSTM achieves stable and accurate prediction of 3D human skeletal joint coordinates and exhibits strong robustness across varying distances (1.5–4 m), with MAE values as low as 0.1075 m (horizontal), 0.0633 m (vertical), and 0.1180 m (depth), significantly outperforming existing radar-based methods such as mmPose, MnPoTr, and M^4^esh. These results demonstrate the practical feasibility of the proposed millimeter-wave radar application for human posture and motion recognition, particularly in some special scenarios such as medical monitoring, smart surveillance, and human–computer interaction.

While the current study focuses on single-person scenarios as a foundational step, future work will expand the dataset to include more complex and challenging cases, such as multi-person interactions and heavily occluded environments, to further enhance the model’s generalization and practicality. Additionally, comparisons with non-radar or hybrid radar-vision methods will be conducted to provide a more comprehensive benchmark and validate the advantages of the radar-based approach. We will also explore graph neural network (GNN) structures for further improving human posture recognition performance.

## Figures and Tables

**Figure 1 sensors-25-05857-f001:**
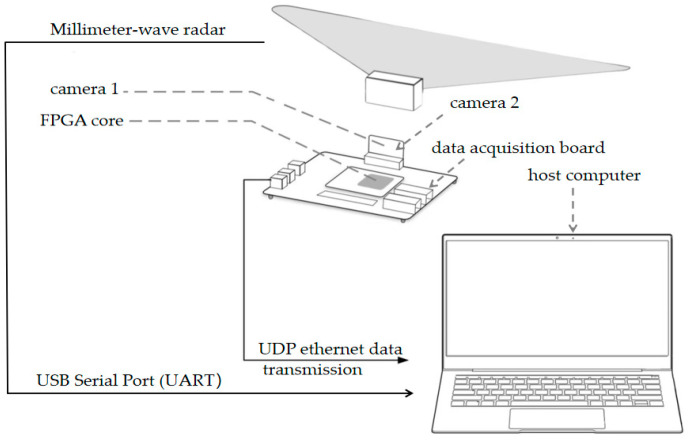
Schematic diagram of hardware connections for the binocular 3D localization and HAR experimental platform.

**Figure 2 sensors-25-05857-f002:**
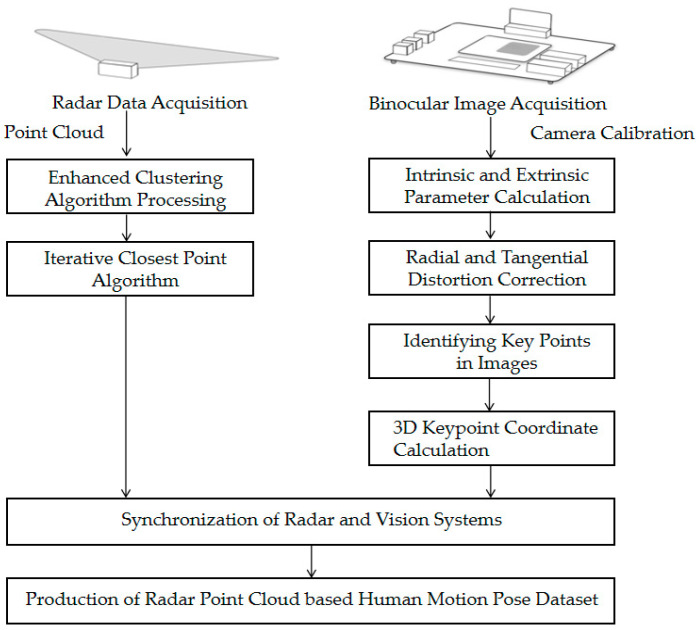
Experimental process for generating a point cloud posture annotation dataset based on binocular vision.

**Figure 3 sensors-25-05857-f003:**
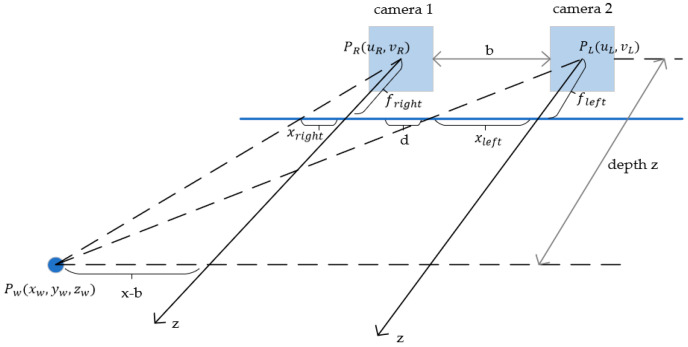
Distance-dimensional projection relationship of binocular cameras.

**Figure 4 sensors-25-05857-f004:**
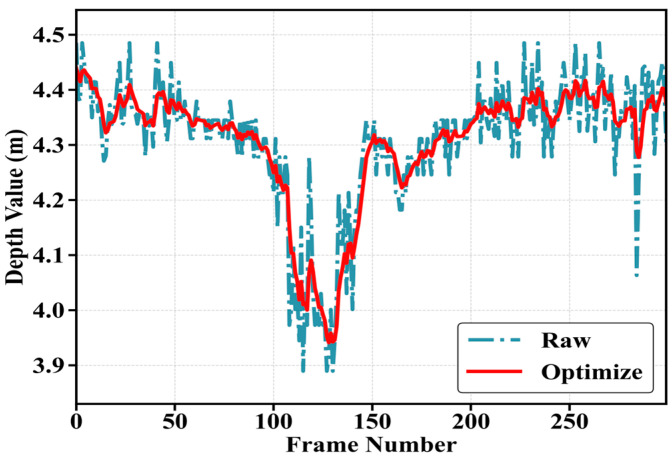
Schematic diagram of the nose keypoints coordinate optimization effect using the Kalman filtering algorithm.

**Figure 5 sensors-25-05857-f005:**
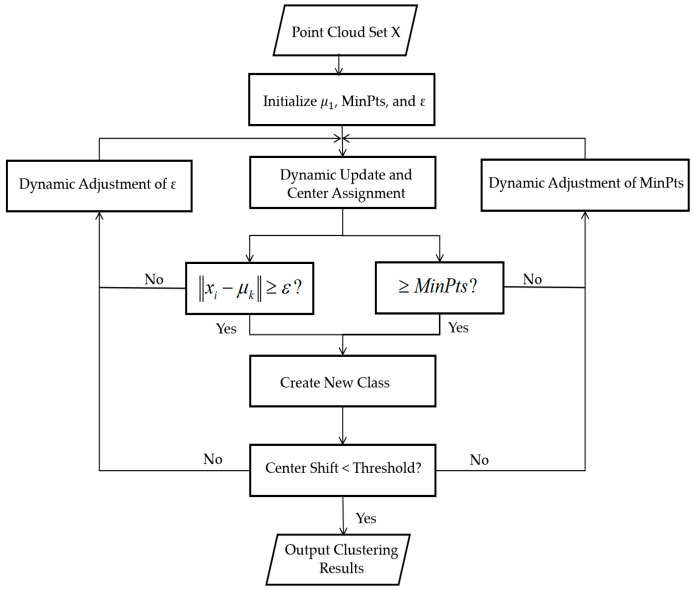
Flowchart of adaptive clustering algorithm based on K-means and DBSCAN fusion.

**Figure 6 sensors-25-05857-f006:**
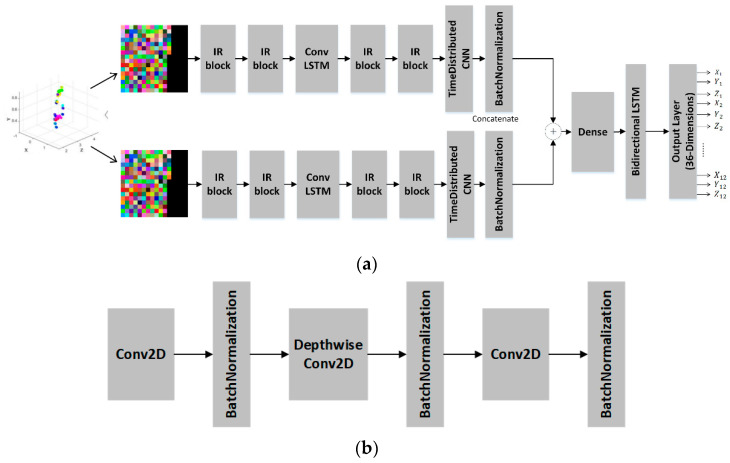
Neural network model architecture diagram for the ST-ConvLSTM. The (*x*, *y*) or (*x*, *z*) coordinates of each point were used as the red and green channel values in the image, while velocity or confidence information was assigned to the blue channel. The output are 3D coordinates for twelve keypoints, with intermediate identical outputs omitted for brevity. (**a**) The overall structure of the ST-ConvLSTM network model; (**b**) the substructure of each IR block.

**Figure 7 sensors-25-05857-f007:**
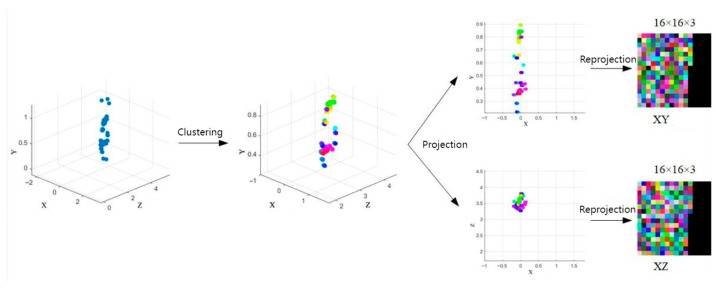
The point cloud data is first processed through the A-DBSCAN clustering algorithm and then projected onto the xoy plane and the xoz plane, respectively. The (*x*, *y*) or (*x*, *z*) coordinates of each point were used as the red and green channel values in the image, while velocity or confidence information was assigned to the blue channel.

**Figure 8 sensors-25-05857-f008:**
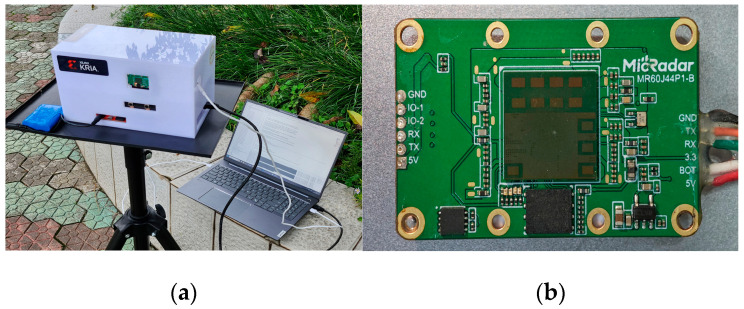
Schematic diagram of the experiment platform. (**a**) Actual data collection using HMAS; (**b**) Millimeter-Wave radar module of the experimental platform.

**Figure 9 sensors-25-05857-f009:**
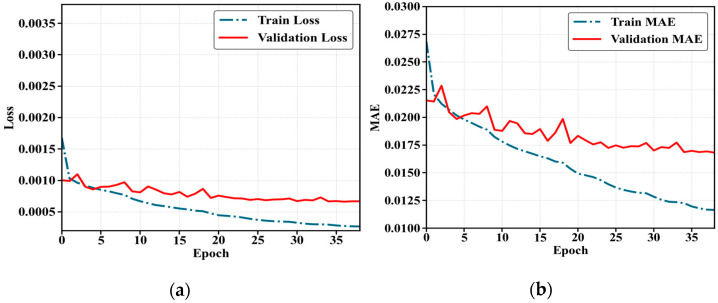
Parameter variations during the model training process. (**a**) The loss curve of the ST-ConvLSTM network training process; (**b**) the MAE curve of the ST-ConvLSTM network training process.

**Figure 10 sensors-25-05857-f010:**
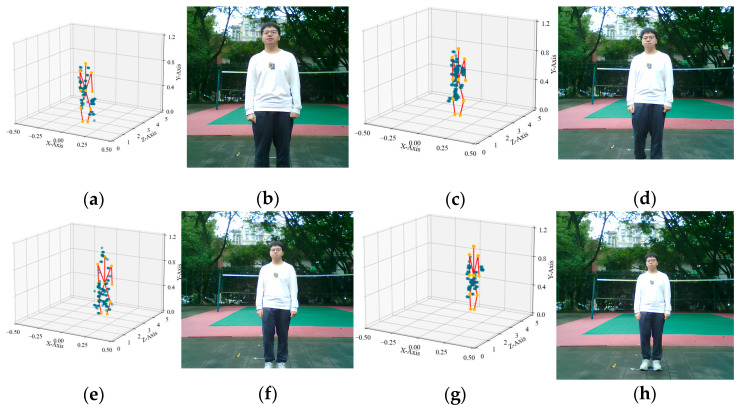
Overlay diagram of model-predicted keypoints and point cloud. The red lines represent the skeletal connections linking the 12 output keypoints, while the blue dots indicate the point cloud data. (**a**) Model output for the frame at *z* = 1.5 m; (**b**) photo corresponding to the frame at *z* = 1.5 m; (**c**) model output for the frame at *z* = 2.5 m; (**d**) photo corresponding to the frame at *z* = 2.5 m; (**e**) model output for the frame at *z* = 3 m; (**f**) photo corresponding to the frame at *z* = 3 m; (**g**) model output for the frame at *z* = 4 m; (**h**) photo corresponding to the frame at *z* = 4 m.

**Figure 11 sensors-25-05857-f011:**
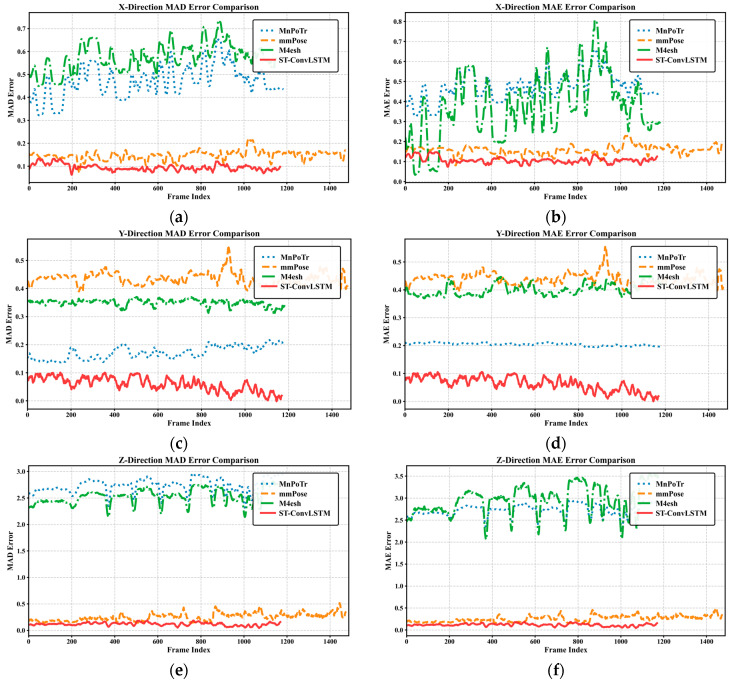
Localization error between model-predicted keypoints and actual coordinates on the training dataset. (**a**,**b**) MAD and MAE curve map in the horizontal direction of the ST-ConvLSTM, mmPose, MnPoTr, and M^4^esh; (**c**,**d**) MAD and MAE curve map in the vertical direction of the ST-ConvLSTM, mmPose, MnPoTr, and M^4^esh; (**e**,**f**) MAD and MAE curve map in the depth direction of the ST-ConvLSTM, mmPose, MnPoTr, and M^4^esh.

**Table 1 sensors-25-05857-t001:** Data composition of the human motion posture dataset.

Dataset Notation	Dataset Description
$K_{t}\in R^{12\times 3}$	3D keypoint coordinates
$P_{t}\in R^{N\times 3}$	point cloud coordinates
$v_{t}\in R^{N\times 1}$	point cloud velocity/SNR
$c_{t}\in R^{N\times 1}$	motion category/confidence
timestamp	current frame timestamp

**Table 2 sensors-25-05857-t002:** Hardware configuration of the computing platform for model training.

Component	Configuration
Central Processing Unit (CPU)	Intel Core i7-8700K (Santa Clara, CA, USA)
Graphics Processing Unit (GPU)	NVIDIA GeForce RTX 2060 (Santa Clara, CA, USA)
System Memory (RAM)	64 GB DDR4
Operating System	Windows 11

**Table 3 sensors-25-05857-t003:** Software environment for model development and training.

Module	Version	Module Function
Python	3.9	High-level programming language
TensorFlow	2.7.0	Neural network backend framework
Keras	2.7.0	High-level neural networks API
OpenCV	3.4.18.65	Computer vision library
NumPy	1.26.4	Scientific computing library

**Table 4 sensors-25-05857-t004:** Ablation study on the contribution of ICP multi-frame fusion. Performance is measured by Mean Absolute Error (MAE) and Median Absolute Deviation (MAD) on the test set (in meters).

Module	MAE	MAD
Single Frame	0.0274	0.0258
ICP Multi-Frame Fusion	0.0115	0.0102

**Table 5 sensors-25-05857-t005:** Statistical analysis of Mean Absolute Error (MAE) curves of the training dataset from Figure 10e–g. Key metrics include mean, range, and median values based on the MAE curve in meters of four network models.

Network Model	Statistic	Horizontal	Vertical	Depth
ST-ConvLSTM	mean	0.1075	0.0633	0.1180
	range	0.1995	0.1945	0.2809
	median	0.1090	0.0648	0.1201
mmPose [21]	mean	0.1576	0.4381	0.2570
	range	0.2358	0.3368	0.9646
	median	0.1584	0.4363	0.2278
MnPoTr [42]	mean	0.4745	0.2044	2.7247
	range	0.3627	0.0360	0.8549
	median	0.4678	0.2046	2.7461
M^4^esh [43]	mean	0.3782	0.4011	2.9600
	range	0.3537	0.2710	0.0466
	median	0.3573	0.3942	3.0033

**Table 6 sensors-25-05857-t006:** The performance of median loss and MAE on the test dataset.

Network Model	Loss	MAE
ST-ConvLSTM	2.6443 × 10^−4^	0.0115
mmPose	8.3952 × 10^−4^	0.0191
MnPoTr	6.8622 × 10^−4^	0.0167
M^4^esh	7.3146 × 10^−4^	0.0172

## Data Availability

The data presented in this study are available upon request from the corresponding authors.

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
