# Peer review of "A ST-ConvLSTM Network for 3D Human Keypoint Localization Using MmWave Radar"

_sensors, 2025, doi:10.3390/s25185857_

Round 1

Reviewer 1 Report

Comments and Suggestions for Authors

The paper provides a radar-based solution for pose estimation. The topic is interesting, and the presentation is generally clear, but there are still some major concerns.

1. The paper does not report the total size and diversity of MRHKD (number of subjects, frames, recording conditions, pose types). This information is necessary to judge whether the reported performance is robust and generalizable.

2 . The paper repeatedly claims high robustness in complex or challenging scenarios. However, Figure 10 shows only a very simple test case (a single, standing person) without examples involving dynamic poses, occlusion, or multiple targets. No qualitative or quantitative comparison with baseline methods is shown for the same visual cases, making it difficult to judge whether ST-ConvLSTM is genuinely more robust or simply performs well in an easy scenario.

3. The pipeline involves multiple stages (A-DBSCAN clustering, ICP multi-frame fusion, dual-projection encoding, ConvLSTM temporal modeling, etc.). Without an ablation study, it is impossible to tell how much of the reported performance gain comes from the ST-ConvLSTM network itself versus the preprocessing/fusion steps. Please provide controlled experiments removing or replacing individual components to clarify their contributions.

4. The choice of mmPose, MnPoTr, and M4esh is reasonable, but no SOTA non-radar or hybrid radar–vision methods are included for comparison. At least one or two such methods should be added to demonstrate the real benefit of the radar-driven approach.

Also, some minor points: 
1. The paper briefly mentions potential extension to medical or other scenarios, but the link is vague. There is already strong evidence that video-based pose estimation works well in medical contexts, e.g.:

a. Zhang, H., Ho, E. S., Zhang, F. X., & Shum, H. P. (2022). Pose-based tremor classification for Parkinson’s disease diagnosis from video. MICCAI, pp. 489–499.
b. Liu, W., Lin, X., Chen, X., Wang, Q., Wang, X., Yang, B., ... & Lin, Y. (2023). Vision-based estimation of MDS-UPDRS scores for quantifying Parkinson’s disease tremor severity. Medical Image Analysis, 85, 102754.

Could you briefly explain how radar-based pose estimation would help or improve upon these existing frameworks? For example, would mmWave provide cleaner, more reliable keypoint tracking in low-light or privacy-sensitive medical settings? Why is that important for these tasks?

2. In the experiments, the GPU (and CPU) hardware configuration should also be reported. Also, please confirm whether the train/validation/test split is subject-exclusive to avoid overestimating performance due to subject-specific overfitting.

3. The paper claims strong potential for wider application but does not explain how easy it would be to deploy the system in different scenarios. Could you elaborate on the practical cost and complexity of transferring this setup to new environments, whether lower-cost radar hardware could be used without significant accuracy loss, and any expected barriers (e.g., installation, calibration, computational requirements) for scaling to multiple sites?

Overall, I recommend a major revision before it can be accepted.

Author Response

Comments 1: The paper does not report the total size and diversity of MRHKD (number of subjects, frames, recording conditions, pose types). This information is necessary to judge whether the reported performance is robust and generalizable.

Response 1: Thank you for pointing this out. We agree with this comment. We have now provided a detailed description of the mmWave Radar Human Keypoints Dataset (MRHKD). The dataset comprises approximately 73,794 frames collected from four subjects in diverse environments, including both indoor and outdoor settings, to incorporate variability in background clutter and lighting conditions. Since our work mainly focuses on the localization of key points in the human body and we have not classified the dataset with pose labels, our dataset does not indicate what types of poses there are. However, the dataset includes various postures of each subject, such as standing, walking, walking forward and backward, turning left and right, raising arms, and tilting left and right. This addition can be found on page 4, paragraph 1.

Comments 2: The paper repeatedly claims high robustness in complex or challenging scenarios. However, Figure 10 shows only a very simple test case (a single, standing person) without examples involving dynamic poses, occlusion, or multiple targets. No qualitative or quantitative comparison with baseline methods is shown for the same visual cases, making it difficult to judge whether ST-ConvLSTM is genuinely more robust or simply performs well in an easy scenario.

Response 2: We sincerely thank the reviewer for this valuable suggestion. Our work primarily focuses on 3D human keypoint localization rather than pose classification or action recognition. Therefore, the qualitative results in Figure 10 are intended to visually demonstrate the model’s ability to accurately regress keypoint coordinates across different distances (1.5–4 m). Quantitative comparisons with baseline methods (mmPose, MnPoTr, M⁴esh) are provided in Figure 11 and Table 5.

Comments 3: The pipeline involves multiple stages (A-DBSCAN clustering, ICP multi-frame fusion, dual-projection encoding, ConvLSTM temporal modeling, etc.). Without an ablation study, it is impossible to tell how much of the reported performance gain comes from the ST-ConvLSTM network itself versus the preprocessing/fusion steps. Please provide controlled experiments removing or replacing individual components to clarify their contributions.

Response 3: We have now included an ablation study on the contribution of the ICP multi-frame fusion module (see Table 4 in Section 6.3). The results show that using ICP fusion reduces the MAE from 0.0274 m (single-frame) to 0.0115 m—a 58% improvement—demonstrating the critical role of temporal integration in enhancing point cloud density and stability. Regarding A-DBSCAN: it is an essential preprocessing step for noise removal and target segmentation, especially in multi-target scenarios, and is not easily ablated without compromising data quality. The dual-projection and ConvLSTM modules are integral to the ST-ConvLSTM architecture and were evaluated as a whole against other end-to-end models. This addition can be found in Section 6.3, pages 16–17.

Comments 4: The choice of mmPose, MnPoTr, and M4esh is reasonable, but no SOTA non-radar or hybrid radar–vision methods are included for comparison. At least one or two such methods should be added to demonstrate the real benefit of the radar-driven approach.

Response 4: We sincerely thank the reviewer for this insightful suggestion. Our mmWave Radar Human Keypoints Dataset (MRHKD) were generated using our Hybrid Human Motion Annotation System (HMAS), which relies on binocular vision and the MoveNet model. Consequently, all models in our study including our ST-ConvLSTM and the baseline models were trained and evaluated on this same radar-specific dataset. This ensures a fair and controlled comparison.

We fully agree that comparing radar-based methods with SOTA vision-based approaches under ideal conditions is valuable—though vision unsurprisingly performs better in such settings. The key advantage of radar lies in challenging scenarios where vision fails, such as low light, occlusions, or privacy-sensitive environments. A meaningful cross-modal comparison would require synchronized radar-vision data under these adverse conditions, which we acknowledge as an important direction for future work. For this paper, our focus was to demonstrate the improvement of our model over existing radar-based methods.

Comments 5: The paper briefly mentions potential extension to medical or other scenarios, but the link is vague. There is already strong evidence that video-based pose estimation works well in medical contexts, e.g.:

  1. Zhang, H., Ho, E. S., Zhang, F. X., & Shum, H. P. (2022). Pose-based tremor classification for Parkinson’s disease diagnosis from video. MICCAI, pp. 489–499.
  2. Liu, W., Lin, X., Chen, X., Wang, Q., Wang, X., Yang, B., ... & Lin, Y. (2023). Vision-based estimation of MDS-UPDRS scores for quantifying Parkinson’s disease tremor severity. Medical Image Analysis, 85, 102754.

Could you briefly explain how radar-based pose estimation would help or improve upon these existing frameworks? For example, would mmWave provide cleaner, more reliable keypoint tracking in low-light or privacy-sensitive medical settings? Why is that important for these tasks?

Response 5: We have expanded the discussion in the Introduction to better articulate the advantages of radar in medical applications. Unlike vision-based systems, mmWave radar is insensitive to lighting conditions, robust to occlusion, and preserves privacy by not capturing identifiable visual features. This makes it particularly suitable for continuous monitoring in home healthcare settings—e.g., detecting falls, abnormal postures, or prolonged inactivity in elderly patients—without intruding on privacy. These attributes address key limitations of video-based systems in real-world medical deployments. This addition can be found in Section 1, pages 2, paragraph 2.

Comments 6: In the experiments, the GPU (and CPU) hardware configuration should also be reported. Also, please confirm whether the train/validation/test split is subject-exclusive to avoid overestimating performance due to subject-specific overfitting.

Response 6: Thank you for pointing this out. We agree with this comment. We have added the hardware configuration used for training in Table 2. This addition can be found in Section 6.1, pages 13.

Additionally, we explicitly state in Section 3.1 that the dataset split is subject-exclusive to prevent subject-specific overfitting and ensure generalizable evaluation. This addition can be found in Section 3.1, pages 4, the last paragraph.

Comments 7: The paper claims strong potential for wider application but does not explain how easy it would be to deploy the system in different scenarios. Could you elaborate on the practical cost and complexity of transferring this setup to new environments, whether lower-cost radar hardware could be used without significant accuracy loss, and any expected barriers (e.g., installation, calibration, computational requirements) for scaling to multiple sites?

Response 7: We have added a new paragraph in Section 6.1 to address this concern. The HMAS system is designed to be compact and portable, consisting of a radar module, binocular cameras, and a computing unit. The system can be recalibrated for new environments using standard stereo calibration procedures. While the 60 GHz radar used here offers high resolution, the architecture is compatible with lower-cost radars as long as they provide basic point cloud output. Real-time inference is efficient and can run on moderate hardware (e.g., embedded NPUs), supporting scalable deployment. Moreover, since the computational load and parameters of our network are not complex, as the computing power of npu becomes increasingly strong, our next research task will also be to transplant the network onto chips equipped with npu for deployment. This addition can be found in Section 6.1, pages 13, the last paragraph.

Reviewer 2 Report

Comments and Suggestions for Authors

The problem of localization of key points of the human body is currently being solved in various ways, but there are enough flaws in this area related to sensitivity, occlusions, and speed. In their work, the authors consider the use of radar to extract the coordinates of body points in three-dimensional space. The use of this equipment leads to the need to solve specific information processing tasks, including the development of an author's model and data markup. The process of marking up radar data based on stereo cameras is described in detail and correctly. The presented approach to converting the source data into two-dimensional arrays and their subsequent reverse convolutional networks seemed promising and justified to me. The practical implementation is described in sufficient detail and includes a set of results in the form of tables and graphs.

My comments are as follows:
- Quantitative results should be added to the Abstract.
- I didn't really like the captions on the legend of Figure 11, although you can understand what models we are talking about, the abbreviations introduced here need to be presented somewhere in the text or description of the figure. 
- It is worth expanding the discussion of the results obtained, based on the results obtained on accuracy and performance, to formulate the scope of the developed solution.

Otherwise, the study is quite detailed, the theoretical and practical parts are fully described.

Author Response

Comments 1: Quantitative results should be added to the Abstract.

Response 1: Thank you for pointing this out. We agree with this comment. We have updated the Abstract to include key quantitative results: MAE values of 0.1075 m (horizontal), 0.0633 m (vertical), and 0.1180 m (depth).  This addition can be found in Abstract, pages 1.

Comments 2: I didn't really like the captions on the legend of Figure 11, although you can understand what models we are talking about, the abbreviations introduced here need to be presented somewhere in the text or description of the figure.

Response 2: We sincerely thank the reviewer for this valuable suggestion. This time, we did not use network abbreviations in the legend but rather the full names of the networks. This addition can be found in Section 6.3, pages 16.

Comments 3: It is worth expanding the discussion of the results obtained, based on the results obtained on accuracy and performance, to formulate the scope of the developed solution.

Response 3: We have expanded the Conclusion section to include a more detailed discussion of the results, highlighting the model's accuracy, robustness, and potential applications in medical monitoring, smart surveillance, and human-computer interaction. We also outline future directions for dataset expansion and model improvement. This addition can be found in Section 7 Conclusion, pages 18–19.

Round 2

Reviewer 1 Report

Comments and Suggestions for Authors

The authors addressed most concerns. While I understand Figure 10 is intended as an illustrative example, more challenging or multi-person cases would be more convincing, and noting such extensions as future work would be useful. Likewise, although non-radar or hybrid SOTA methods are not included, acknowledging this as a future direction would also strengthen the paper.

Author Response

Comments: While I understand Figure 10 is intended as an illustrative example, more challenging or multi-person cases would be more convincing, and noting such extensions as future work would be useful. Likewise, although non-radar or hybrid SOTA methods are not included, acknowledging this as a future direction would also strengthen the paper.

Response: Thank you for this constructive suggestion. We fully agree that including more challenging scenarios such as multi-person interactions would further demonstrate the robustness of the proposed method. In response to your comment, we have explicitly acknowledged this limitation and outlined it as a direction for future work. This addition can be found in Section 7 Conclusion, pages19, paragraph 2.

  Additionally, we also agree that comparisons with non-radar or hybrid radar–vision methods would provide a more comprehensive evaluation of the radar-based approach. Accordingly, we have added extending this work toward cross-modal comparisons in the future. This addition can be found in Section 7 Conclusion, pages19, paragraph 2.